# A Multi-Objective Optimization Problem for Optimal Site Selection of Wind Turbines for Reduce Losses and Improve Voltage Profile of Distribution Grids



**Amirreza Naderipour [1], Zulkurnain Abdul-Malek [1,\*], Saber Arabi Nowdeh [2], Foad H. Gandoman [3,4] and Mohammad Jafar Hadidian Moghaddam [5]**

[1] Institute of High Voltage & High Current, Faculty of Electrical Engineering, Universiti Teknologi Malaysia, Johor 81310, Malaysia

[2] Electrical Department, Payambar′azam Student Research Center, Golestan 87349-49318, Iran

[3] Research Group MOBI—Mobility, Logistics, and Automotive Technology Research Center, Vrije Universiteit Brussel, Pleinlaan 2, 1050 Brussels, Belgium

[4] Flanders Make, 3001 Heverlee, Belgium

[5] College of Engineering and Science, Victoria University, Melbourne 3047, Australia

\* Correspondence: zulkurnain@utm.my

**Abstract:** In this paper, the optimal site and size selection of wind turbines (WTs) is presented considering the maximum allowable capacity constraint with the objective of loss reduction and voltage profile improvement of distribution grids based on particle swarm optimization (PSO as a multi-objective problem using weighted coefficients method. The optimal site, size, and power factor of the WTs are determined using PSO. The proposed method is implemented on 84- and 32-bus standard grids. In this study, PSO algorithm is applied to determine the size, site, and power factor of WTs considering their maximum size constraint (with constraint, variant size) and also not considering their maximum size constraint (without constraint, constant size). The simulation results showed that the PSO is effective to find the site, size, and power factor of WTs optimally in the single and multi-objective problem. The results of this method showed that the power loss is reduced more and voltage profile improved more considering WTs maximum allowable size versus not considering this constraint. Additionally, the multi-objective results showed that there is a compromise between the objectives in the multi-objective WTs site selection and the multi-objective problem solution is a more realistic and accurate approach in comparison with the single-objective problem solution.

**Keywords:** distribution grid; loss reduction; improving voltage profile; maximum allowable wind turbine capacity; Particle Swarm Optimization

## 1. Introduction

Owing to climate changes, such as global warming and increased $CO_2$ emissions, there is an urgent requirement for power production based on distributed power generation sources (DPGSs). One such concept is to generate electricity closer to the customer, which is known as distributed energy generation. Generating energy closer to the load reduces the requirement for long distance power lines. However, establishing a reliable connection between the DPGSs and the utility grid may be a challenge [1]. The attention on DPGSs, such as wind turbine (WT), solar, geothermal, and fuel cells are constantly increasing with the growing requirements for new clean and sustainable technologies. Power generation by a WT demonstrates advantages such as reduction in grid power losses, improvement in voltage profile, improvement in peak traffic, and faster transportation on transmission and distribution lines [2]. In contrast, the reduction in power quality, reduction in

reliability, and voltage and protection problems are certain disadvantages of integrating WTs with a distribution system, which may occur if the size and site of WTs are not appropriate [3]. The incorrect selection of site, number, and/or capacity of the WT results in distribution system problems such as voltage deviation and power losses. Therefore, determining the optimal amount and location of WTs is very important [4]. In Reference [5], the optimal use of distributed generation resources has been used to improve the reliability of the system, reduce losses, improve the voltage profile, and the genetic algorithm (GA) is applied to solve the optimization problem. In Reference [6], based on the tabu search (TS) algorithm, Loss reduction of distribution system using DGs has been investigated. In Reference [7], an artificial bee colony (ABC) algorithm is used to minimize power losses based on the determination of optimal distributed generation location and size. In Reference [8], the location of DGs is aimed at reducing active power losses and improving the voltage profile to improve network performance. For this purpose, the backtracking search optimization algorithm (BSOA) has been used along with a set of fuzzy expert rules. In Reference [9], wind turbines placement is presented with objective of net profit maximizing using GA. In Reference [10], the location and number of wind turbines in a wind farm were aimed at minimizing generation costs. In this paper, the type of wind turbine has also been studied in the optimization problem. In Reference [11], the optimal determination of wind turbines location was considered with regard to constant speed and no change in wind direction. In Reference [12], the optimal location of wind turbines has been used to increase the generation level and reduce the land occupied by the wind farm using GA. In Reference [13], a method for optimal placement wind turbines was presented in a radial distribution system based on the evolutionary programming (EP) method. In Reference [14], the optimization of the location and size of the combination of solar panels and wind turbines in the distribution network is proposed using the ant lion optimizer (ALO) with the goal of reducing losses and improving the profile and voltage stability. In Reference [15], the PSO algorithm has been used separately for the installation of wind turbines and photovoltaic panels. The problem is expressed as bound and has a multi-criteria objective function, which is to minimize losses and improve voltage stability. In Reference [16], the location of wind turbines and solar panels has been proposed to reduce power losses and improve voltage stability using PSO.

According to the mentioned studies the most placement researches were done considering the variable size of the WTs. A two-step particle swarm optimization (PSO) algorithm to facilitate the placement of WT generations considering their maximum allowable capacity with the objective of loss reduction in distribution system as single objective is proposed in Reference [17]. Hence, the control variables were obtained and for the optimized control variables, the maximum output of WTGs was changed based on the minimization of power losses. In Reference [17], the impact of constant and variant maximum size of WTs is evaluated, but with single objective as the loss reduction of the distribution system.

In this paper, a method for optimal site selection of WTs is provided while considering their maximum allowable capacity with objective of losses reduction and voltage profile improvement in distribution system using PSO algorithm as multi-objective optimization problem considering constant (the size of WTs were considered constant, without considering size constraint) and variant (the size of WTs were considered variant as variable, with considering size constraint) maximum size of WTs. Initially, the problem statement and the objective function of optimization with its constraints are presented. The weight factors method is applied for multi-objective optimization solution. The PSO algorithm is used to determine the optimal site and size of wind turbines in the distribution systems. Besides, in this study, the WTs placement problem is presented in different numbers of turbines with single and multi-objective optimization with and without regard to the maximum allowable capacity of WTs.

## 2. Statement of the Problem

### 2.1. Objective Function

In this paper, the objective function of the site selection of WTs in the distribution grid is considered as minimizing losses and improving the voltage profile. The objective function of minimizing active power losses is presented below [6].

$$F_1 = P_{loss} = \sum_{k=1}^{N_{br}} R_k I_k^2 \tag{1}$$

$$I_k = \frac{V_j - V_i}{R_k + jX_k} \tag{2}$$

where $k$ is the line number between the bus $i$ and $j$, $R$ and $X$ are the line resistance and reactance, $I$ is the current passing through the line and $P_{loss}$ is the line losses.

The objective function of improving the voltage profile is defined as follows:

$$F_2 = \sum_{n=1}^{N_{bus}} |V_m(n) - 1| \tag{3}$$

where $V_m(n)$ is the bus of $i$ voltage and $N_{bus}$ is the number of buses of the distribution grid. In this study, optimization variables include WT's site and power factor in the grid. Decision vector X has two sections. The first section shows the installation site of the WT's generation unit in the grid. The second section of the vector X is the power factor [6].

$$X = [P_{WTG} pf_{WTG}] \tag{4}$$

The objective function is formulated based on weighted coefficients method as $F = W_1 (F_1/F_{1,max}) + W_2 (F_2/F_{2,max})$. The $W_1$ (weighted coefficient of $F_1$) and $W_2$ (weighted coefficient of $F_2$) are selected 0.6 and 0.4 (based on user experience), respectively. $F_{1,max}$ and $F_{2,max}$ is maximum value of $F_1$ and $F_2$, respectively.

### 2.2. Constraints

The objective function of the problem should be optimized under the following constraints:

- Power of the distribution line

$$\left|P_{ij}^{Line}\right| < P_{ij,max}^{Line} \tag{5}$$

where $P_{ij}^{Line}$ is the power of the line and $P_{ij,max}^{Line}$ is the thermal limit of the line.

- Load distribution equations

$$P_i = \sum_{i=1}^{N_{bus}} V_i V_j V_{ij} \cos(\theta_{ij} - \delta_i + \delta_j), Q_i = \sum_{i=1}^{N_{bus}} V_i V_j V_{ij} \sin(\theta_{ij} - \delta_i + \delta_j) \tag{6}$$

where $P_i$ and $Q_i$ are the injected active and reactive powers, $V_i$ and $\delta_i$ are the bus $i$ voltage domain and angle. $Y_{ij}$ and $\theta_{ij}$ are the domain and admittance angle of the ramification between bus $i$ and $j$.

- Lines' loading

$$\left|L_-(f,i)\right| < L_-(f,i) max i = 1, 2, \ldots, N_-f \tag{7}$$

where $Nf$ is the number of feeders, $L_{f,i}$ and $L_{f,i}^{max}$ are the domain and maximum current of feeder $i$.

- Maximum reactive power of the WT

$$P_{min,w,i} \leq P_{w,i} \leq P_{max,w,i} \tag{8}$$

where $P_{min}$, $w$, $i$ and $P_{max}$, $w$, $i$ are the minimum and maximum authorized power of WTG$_i$.

- Bus voltage

$$V_{min} \leq V_i \leq V_{max} \tag{9}$$

where $V_{min}$ and $V_{max}$ are the minimum and maximum bus voltages.

- WT's power factor

$$pf_{min,i} \leq pf_i \leq pf_{max,i} \tag{10}$$

where $pf_{min}$,$i$, and $pf_{max}$,$i$ are the minimum and maximum values of the power factor WTG$_i$.

## 3. Solving Method

### 3.1. Particle Swarm Optimization

Particle Swarm Optimization (PSO) has been used as one of the most powerful methods for solving optimization problems that have high convergence speed and accuracy in power system studies. On the basis of PSO, the population is called swarm and each member of the population is called particle [18,19]. Particles move in the n-dimensional search space (*n* is the number of optimization variables). Each particle can be a possible answer to the optimization problem. Since each particle is introduced only with its position and velocity, the mathematical model of this algorithm can be expressed as follows:

$$S_i(t) = (S_{i,1}(t), S_{i,2}(t), \ldots, S_{i,n}(t)) \tag{11}$$

where $S$ is an n-dimensional vector representing the position of the element $i$ in the time $t$ and $n$ is the number of optimization variables. For example, $S_{(i,1)}(t)$ indicates the state (value) of the first optimization dimension (the first variable) of particle $i$ at the moment $t$. The velocity of each particle at $t$ is given by the following equation [18].

$$V_i(t) = (V_{i,1}(t), V_{i,2}(t), \ldots, V_{i,n}(t)) \tag{12}$$

In fact, relation (12) represents the velocity of changing the position of an element. For example, $V_{i,1}(t)$ shows the velocity of optimizing the particle $i$ to the first dimension (first variable) at $t$. At the beginning, the particles are created with random sites and velocities, and then, according to the following relationships, each particle is updated using the two best values [18].

$$P_i(t) = (P_{i,1}(t), P_{i,2}(t), \ldots, P_{i,n}(t)) \tag{13}$$

$$P_{gb}(t) = \left(S_{gb,1}(t), S_{gb,2}(t), \ldots, S_{gb,n}(t)\right) \tag{14}$$

where $P$ is an n-dimensional vector, which shows the best position of a particle until t. $P_{gb}$ is also an n-dimensional vector, which shows the best position in the entire community until $t$ (the best situation for one of the particles in the group). It should be noted that $P_i(t)$ and $P_{gb}(t)$ are updated in each $t$ period. Changes in time at $t + 1$ (changes in the site and rate of position change of each particle) are defined as relations (15) and (16). In the process of this algorithm and in each repetition of the process, each member remembers its best position and the best position in the entire community and proceeds to change its position based on these two.

$$S_i(t+1) = k\left[w(t)(t)S_i(t) + C_1 rand_1(P_i(t) - S_i(t)) + C_2 rand_2\left(P_{gb}(t) - S_i(t)\right)\right] \tag{15}$$

$$S_i(t+1) = S_i(t) + V_i(t+1) \tag{16}$$

In Equation (15), $C_1$ and $C_2$ are the power factors between two community and individual forces. In other words, as the $C_1$ is greater than $C_2$, the individual's influence from the community is less and acts individually and as it is lower, it is more influenced by the community. In a real community, since not all the elements are similar, these factors are different. *W(t)* is the factor of inertia (the lack of desire to change the path) in a particle, and rand1 and $rand_2$ are random numbers distributed with a uniform distribution between zero and one. With respect to Equation (16), it can be concluded that each particle, when moving, takes into account the best position it has experienced in its previous movement and considers the best position for the whole group. The coefficient k in Equation (15) is used to ensure convergence and is defined as Equation (17). To ensure convergence, its best value is $k = 0.75$. Depending on the definition of the $C_1$ and $C_2$ coefficients as well as *w*, different versions of the PSO algorithm are created. *w(t)* is to control the diversity and variety of exploration (to reach different and possible solutions in the problem space) and particle convergence. In order to avoid divergence, it is necessary that elements search the space with smaller steps over time; therefore, in the improved PSO method, each coefficient w changes in each repetition. At the beginning, this parameter is set to the largest value of $w_{max}$ to expand the scope of the exploration in the problem space and then (to achieve a more accurate answer) decreases linearly (with a constant gradient) until reaching $w_{min}$ in the last iteration and its size is calculated by Equation (18) for each iteration [18].

$$k = \frac{2}{\left|2 - \varphi - \sqrt{\varphi^2 - 4\varphi}\right|}, \varphi = C_1 + C_2, \varphi \geq 4 \tag{17}$$

$$w(t) = w_{max} - \frac{w_{max} - w_{min}}{t_{max}} t \tag{18}$$

In Equation (18), $t_{max}$ is the maximum iteration and $t$ is the number of iterations at the same time.

In this study the decision variables include of optimal size, site and power factor of WTs are determined using PSO and are described in detail in the next section. In this study, the assumptions of the problem are that the maximum size of each wind turbine for 84-bus grid is considered 5 MW and for 32 bus grid each wind turbine size is considered 2 MW. Operating constraints are also considered, and the minimum and maximum bus voltages are considered between 0.9 and 1.05 p.u. Additionally, the $C_1$ and $C_2$ parameters for the PSO are considered equal to 2, also $w_{max}$ and $w_{min}$ are equal to 0.9 and 0.4, which are selected based on the trial and error method.

### 3.2. The Proposed Method

There are two steps to this method. In the first step, control variables are determined. In the second step, the power output of WTs is based on the proposed objective function and the observation of the problem constraints for the obtained control variables. During these two stages, the installation site, power factor, and maximum capacity of the WT are determined. In the first stage, the installation site and the power factor of the WTs are determined and in the second stage, their maximum capacity is determined. The problem-solving procedure is as follows:

(Step 1–1) Random generation of the initial population.

(Step 2–1) To calculate the maximum turbine capacity, scenarios are created according to the number of turbines, i.e., 2n_1 (n refers to the number of WTs). For example, if there are 4 Turbines, there will be 15 scenarios that, by dividing them by 4 cases, the capacity of WTs will change to satisfy the constraints.

(Case 1) Capacity of three turbines is constant and that of the other one is variable.

(Case 2) Capacity of two turbines is constant and that of the other two is variable.

(Case 2) Capacity of one turbine is constant and that of the other three is variable.

(Case 2) Capacity of all four turbines is variable.

(Step 2–2) The maximum capacity of the WTs for each scenario in the previous step is determined as follows.

If the grid voltage deviation is exceeded, increase the capacity of the WTs so that the voltage deviation is within the permissible range.

If the standard deviation of the grid voltage is less than the optimal value and the loading of the lines is excessive, reduce the capacity of the WTs put the loading of the lines within the allowable range.

The above steps are implemented for all stages and the results of all scenarios are saved based on the proposed objective function.

(Step 2–3) The scenario is extracted with the lowest amount of losses corresponding to the control variables in step 1 among the different scenarios.

(Step 1–2) The primary particle population is sorted according to the objective function.

(Step 1–3) Gbest and Pbest is extracted from the sorted population.

(Step 1–4) The acceleration of the particles is updated.

(Step 1–5) The position of the particles is updated at this stage.

(Step 1–6) The initial population of particles is sorted according to the evaluated objective function.

(Step 1–7) Gbest and Pbest are extracted from the updated population.

(Step 1–8) If convergence conditions are met, the program stops, otherwise go to step 2-1.

The flowchart of the problem solution is shown in Figure 1.

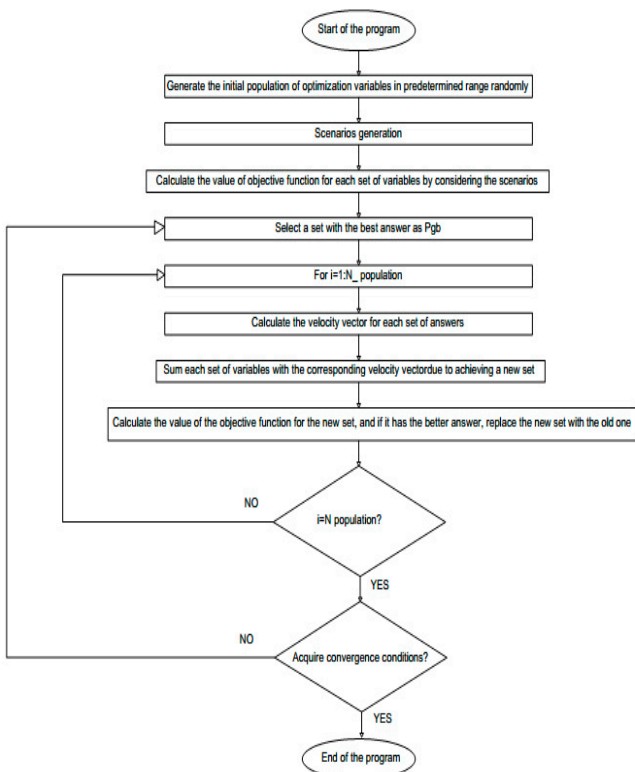

**Figure 1.** The flowchart of the problem solution.

## 4. Simulation Results and Discussion

In this section the results of optimal site selection of wind turbines in 84 and 32 bus distribution systems are presented as single and multi-objective optimization with and without maximum allowable capacity of the WTs using the PSO algorithm. Wind turbines generate the power with receiving the wind speeds. In this study, it is assumed that the maximum size of each wind turbine, receiving the wind speeds for 84 bus system, is 5 MW and for 32-bus system, is 2 MW, considering the wind speed of that region. Therefore, every wind turbine is considered in terms of its peak generation capacity [19]. In the placement problem, the site, size as well as the power factor of each wind turbine is optimally determined by the optimization algorithm.

### 4.1. 84-Bus Grid

The studied grid is a 84-bus IEEE grid the single-line diagram of which is shown in Figure 2. The grid is divided into 4 zones and 4 WTs are in four zones. In other words, a WT is considered for each zone. The 84-bus grid has active and reactive loads of 28,350 and 20,700 kW, respectively. The total loss of this grid is 531.8 kW and the lowest voltage of this grid occurs at bus 10, which is 0.9285. The information on this grid is taken from Reference [17]. In this section, the maximum capacity of each WT is considered 5 megawatts.

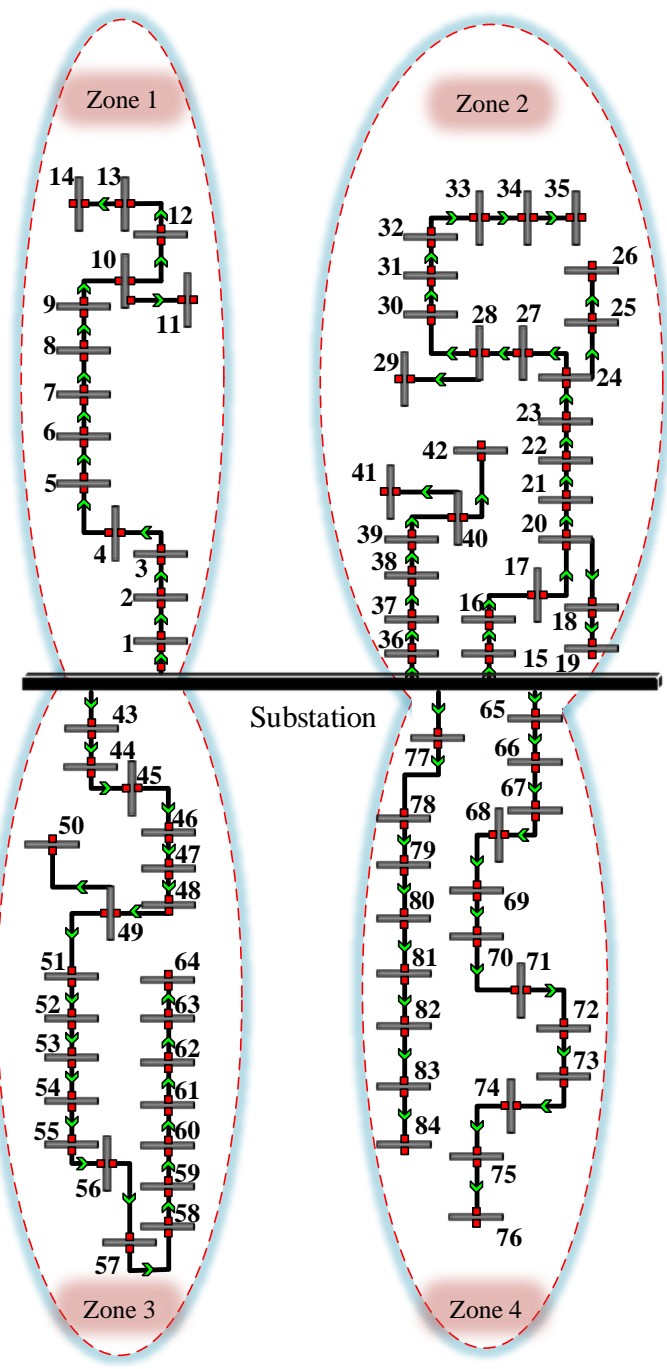

**Figure 2.** Schematics of the 84-bus grid.

### 4.1.1. Optimal Site Selection of Turbines Without Regard to Constraints

Initially, without considering the constraints of the problem, the site selection of the turbines in done for a few cases to examine the effect of increasing the number of turbines on the objective function. At first, the problem is solved as single-objective to reduce the losses and then as multi-objective based on the weight factor method. In this study, the population of the algorithm is 30 and the maximum number of iterating the algorithm is 100.

- **Results of solving the problem as single-objective**

The convergence curve of the PSO method in single-objective optimization is presented in Figure 3. Accordingly, with the increase in the number of WTs, the value of the objective function improves and fewer losses occur. The results of these optimizations are presented in Table 1.

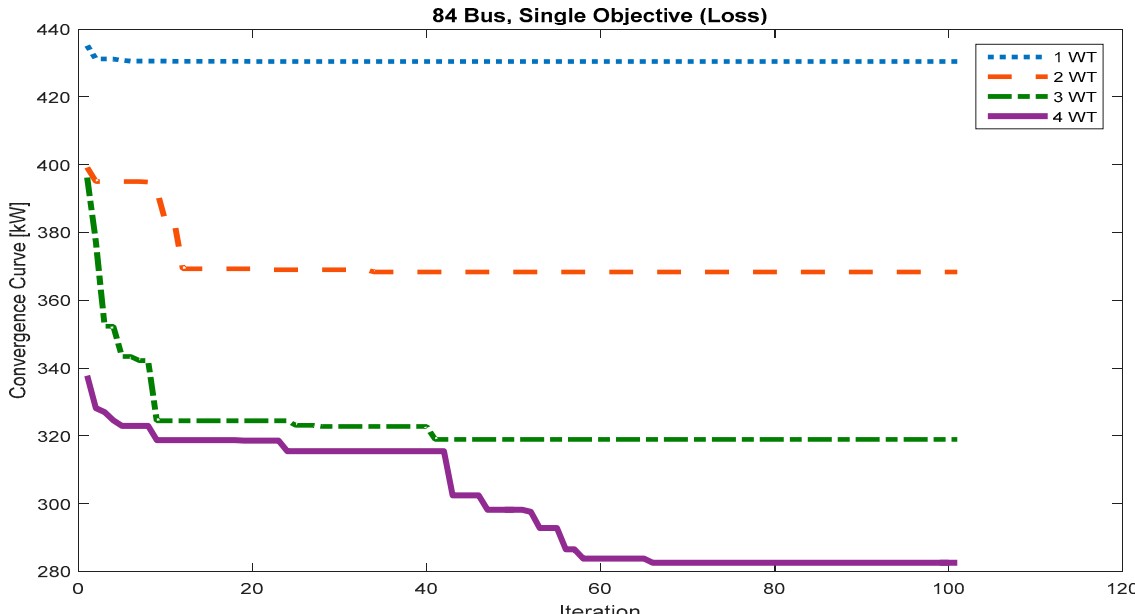

**Figure 3.** Convergence curve of the PSO method in the optimal site selection of turbines without considering constraints (single-objective).

**Table 1.** This Results of optimal turbine site selection without considering constraints (single-objective).

| Number of WTG | WTG Placement | | | | Loss | Percent Reduction of Loss | Minimum Voltage | Maximum Voltage |
|---|---|---|---|---|---|---|---|---|
| 0 | | - | | | 531.8 | 0 | 0.9258 | 1.00 |
| 1 | Bus | | 8 | | 430.45 | 19.05 | 0.9286 | 1.00 |
| | Rate (kW) | | 5000 | | | | | |
| | PF | | 0.9065 | | | | | |
| 2 | Bus | | 8 | 81 | 368.31 | 30.74 | 0.9488 | 1.015 |
| | Rate (kW) | | 5000 | 5000 | | | | |
| | PF | | 0.9037 | 0.8872 | | | | |
| 3 | Bus | 8 | 33 | 81 | 318.93 | 40.02 | 0.9488 | 1.013 |
| | Rate (kW) | 5000 | 5000 | 5000 | | | | |
| | PF | 0.9117 | 0.8828 | 0.8937 | | | | |
| 4 | Bus | 8 | 21 | 33 | 81 | 282.54 | 46.87 | 0.9488 | 1.017 |
| | Rate (kW) | 5000 | 5000 | 5000 | 5000 | | | | |
| | PF | 0.8952 | 0.9429 | 0.8828 | 0.8579 | | | | |

As is known, the increase in the number of turbines has had a great effect on reducing losses and grid voltage. In addition, the grid voltage profile with WTs is shown in Figure 4, which indicates that, with increasing their number, the voltage oscillations decrease, and the grid voltage profile is improved. It should be noted that the results of single-objective optimization, including losses and minimum and maximum voltages, are consistent with the results of Reference [17].

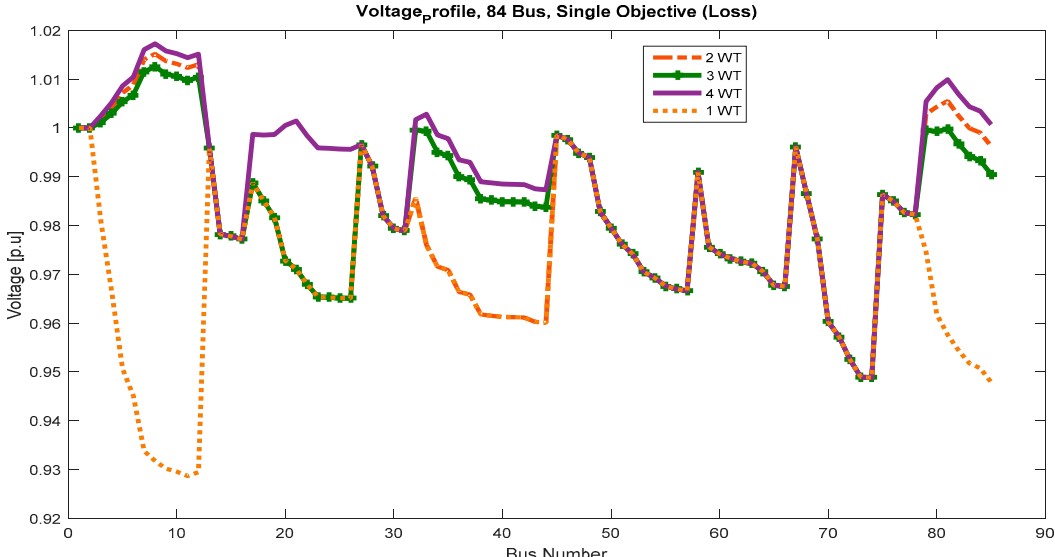

**Figure 4.** This Convergence 84-bus grid voltage profile with and without using WTs without regard to constraints (single-objective).

- **Results of solving the problem as multi-objective**

The results of the optimal site selection of turbines without considering the constraints (multi-objective) are presented in Table 2. According to the results, it has been observed that using optimal WTs in the distribution grid and increasing their number, grid losses are significantly decreased, and grid voltage values are improved. The 84-bus voltage profile curve for multi-objective optimization is also presented in Figure 5, which shows that, with increasing number of WTs, the voltage profile is relatively better.

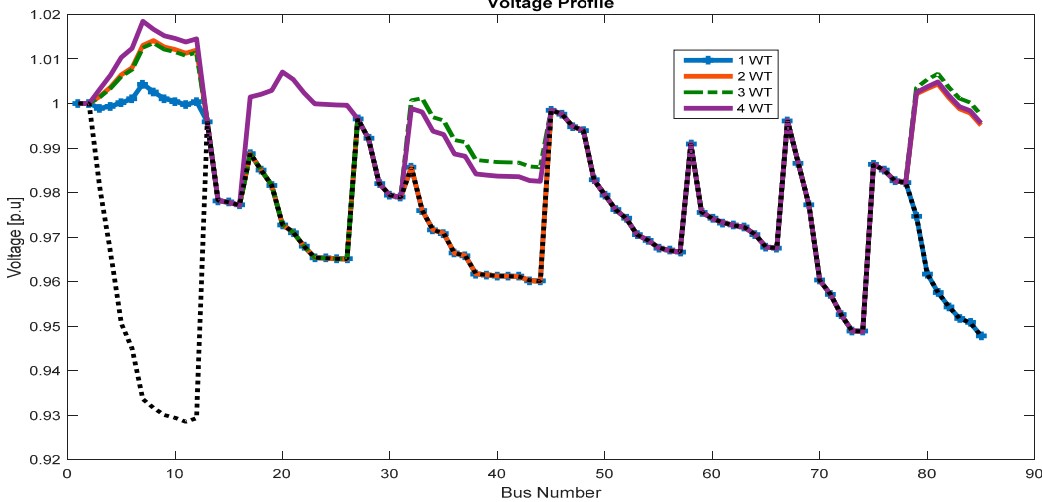

**Figure 5.** 84-bus grid voltage profile with and without using WTs without regard to constraints (multi-objective).

**Table 2.** Results of optimal site selection of turbines without considering constraints (multi-objective).

| Number of WTG | WTG Placement | | | | Loss | Percent Reduction of Loss | Minimum Voltage | Maximum Voltage |
|---|---|---|---|---|---|---|---|---|
| 0 | | - | | | 531.8 | 0 | 0.9285 | 1 |
| 1 | Bus | | 7 | | 433.26 | 18.52 | 0.9478 | 1.004 |
| | Rate (kW) | | 5000 | | | | | |
| | PF | | 0.941 | | | | | |
| 2 | Bus | 8 | | 81 | 368.4 | 30.72 | 0.9488 | 1.016 |
| | Rate (kW) | 5000 | | 5000 | | | | |
| | PF | 0.9037 | | 0.8872 | | | | |
| 3 | Bus | 9 | 33 | 81 | 326.72 | 38.56 | 0.9488 | 1.014 |
| | Rate (kW) | 5000 | 5000 | 5000 | | | | |
| | PF | 0.9168 | 0.9529 | 0.9003 | | | | |
| 4 | Bus | 7 / 20 | 33 | 81 | 284.5 | 46.5 | 0.9488 | 1.018 |
| | Rate (kW) | 5000 / 5000 | 5000 | 5000 | | | | |
| | PF | 0.8845 / 0.8804 | 0.9445 | 0.8914 | | | | |

In Table 3, the comparison of the losses and single and multi-objective voltage optimization are presented in 84-bus grid. Accordingly, in single-objective optimization, all attention is paid to reducing losses, but when the voltage profile is also considered as part of the objective function, the proposed method for minimizing voltage oscillations of the grid buses is considered as well. Therefore, grid losses in multi-objective optimization are slightly higher than its value in the single-objective optimization. On the other hand, the maximum voltage values in a multi-objective mode are slightly better than the single-objective mode.

**Table 3.** Comparison of losses and single and multi-objective optimization voltages.

| Title | Loss | | Minimum Voltage | | Maximum Voltage | |
|---|---|---|---|---|---|---|
| Optimization | Single Objective | Multi-Objective | Single Objective | Multi-Objective | Single Objective | Multi-Objective |
| One WT | 430/45 | 433/26 | 0/9286 | 0/94,785 | 1 | 1/004 |
| Two WT | 368/31 | 368/31 | 0/9488 | 0/9488 | 1/015 | 1/016 |
| Three WT | 318/93 | 326/72 | 0/9488 | 0/9488 | 1/013 | 1/014 |
| Four WT | 54/282 | 284/50 | 0/9488 | 0/9488 | 1/017 | 1/0184 |

### 4.1.2. Optimal Site Selection of Turbines with Respect to Constraints

Here, it is attempted to select the suitable site for the installation of four WTs with respect to constraints. For this purpose, as shown in Figure 2, the grid is divided into four zones. Due to the presence of 4 WTs, 15 scenarios have been created that are categorized in four different cases as stated. It is necessary to note that in this section, the problem solution is presented as a multi-objective. The results of the optimization are presented in Table 4. Accordingly, the variable considering all WTs in the fourth case leads to the highest losses reduction and improves grid voltage profiles. The grid voltage profile before and after the installation of WTs is shown in Figure 6. It is clear that the presence of WTs has greatly improved the grid voltage profile.

**Table 4.** Results of WT site selection for84-bus grid with respect to constraints.

| Number of WTG | | WTG Placement | | | | Loss | Percent Reduction of Loss | Minimum Voltage | Maximum Voltage |
|---|---|---|---|---|---|---|---|---|---|
| 1 | Bus | 8 | 33 | 52 | 81 | 289.71 | 45.52 | 0.9488 | 1.011 |
| | Rate (kW) | 4130.84 | 5000 | 5000 | 5000 | | | | |
| | PF | 0.9395 | 0.9404 | 0.9049 | 0.9271 | | | | |
| 2 | Bus | 8 | 33 | 54 | 81 | 274.75 | 48.33 | 0.9488 | 1.007 |
| | Rate (kW) | 4130.8 | 5000 | 4130.8 | 5000 | | | | |
| | PF | 0.9182 | 0.947 | 0.9156 | 0.8889 | | | | |
| 3 | Bus | 8 | 33 | 54 | 81 | 266.04 | 49.97 | 0.9488 | 1.0052 |
| | Rate (kW) | 4130.8 | 5000 | 4130.8 | 4130.8 | | | | |
| | PF | 0.9055 | 0.9252 | 0.9284 | 0.9121 | | | | |
| 4 | Bus | 8 | 33 | 54 | 81 | 262.95 | 50.55 | 0.9488 | 1.017 |
| | Rate (kW) | 4130.8 | 4130.8 | 4130.8 | 4130.8 | | | | |
| | PF | 0.9033 | 0.9309 | 0.8847 | 0.9023 | | | | |

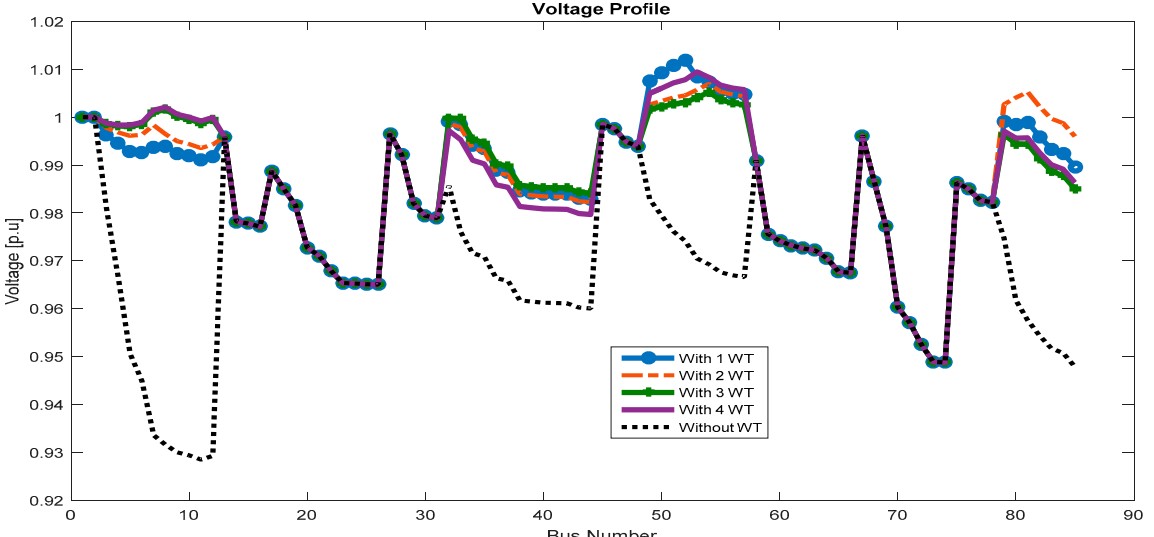

**Figure 6.** 84-bus grid voltage profile before and after installing the turbines with respect to constraints.

In Table 5, the results are compared with and without regard to the constraints in the solution of the 84-bus grid problem. As is clear, taking into account the constraints of the problem leads to a greater loss reduction. On the other hand, with respect to the constraints, but the min voltages are lower than the case without regard to the constraints.

**Table 5.** Comparison of results with and without regard to the 84-bus grid.

| | Loss | | Minimum Voltage | | Maximum Voltage | |
|---|---|---|---|---|---|---|
| Optimization | With Constraints | Without Constraints | With Constraints | Without Constraints | With Constraints | Without Constraints |
| One WT | 289.71 | 433.26 | 0.9488 | 0.94785 | 1.011 | 1.004 |
| Two WT | 274.75 | 368.31 | 0.9488 | 0.9488 | 1.007 | 1.016 |
| Three WT | 266.04 | 326.72 | 0.9488 | 0.9488 | 1.0052 | 1.014 |
| Four WT | 262.95 | 284.50 | 0.9488 | 0.9488 | 1.017 | 1.0184 |

### 4.1.3. 32-Bus Grid

The second grid is a 32-bus grid that its single-line diagram is depicted as two separate zones in Figure 7. The information on this grid is taken from Reference [17]. In this section, the maximum capacity of each WT is considered 2 megawatts. The 32-bus grid in normal conditions has an active loss of 202.67 kW.

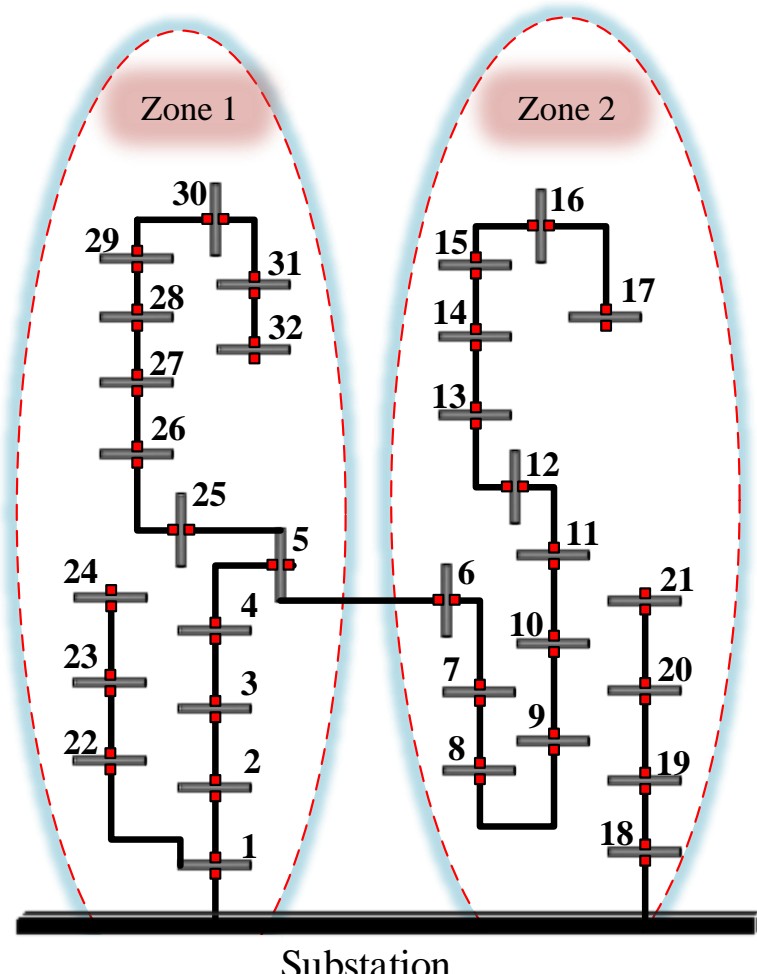

**Figure 7.** The single-line diagram of IEEE 32-bus grid.

Two WTs are considered for this grid that their site should be selected by the proposed method. In this section, two simulation models are considered. First, one of the turbines is considered as constant at its maximum capacity and the second turbine capacity is selected as variable. In the second simulation, the capacity of both turbines is considered variable. The results of the optimizations are presented in Table 6. According to Table 6, the losses in the second case are reduced, which is due to the variable capacity of both WTs. In the first case, where the capacity of the turbines is constant, and in the case that the capacity of both of them is variable, the losses of the 32-bus grid are reduced by 72.7% and 73.14%, respectively. In addition, the minimum and maximum voltages are better in the second case. The grid voltage profile of 32-bus grid before and after the installation of WTs is shown in Figure 8. The presence of WTs has greatly improved the grid voltage profile. In addition, in the second case and in the case that the capacity of both turbines is variable, the voltage oscillations of the grid are less than the case that both turbines are constant.

**Table 6.** Results of site selection for 32-bus grid with respect to constraints.

| Condition | WTG Placement | | | Loss | Percent Reduction of Loss | Minimum Voltage | Maximum Voltage |
|---|---|---|---|---|---|---|---|
| 1 | Bus | 7 | 24 | 55.34 | 72.7 | 0.9654 | 1.008 |
| | Rate (kW) | 2000 | 1625.33 | | | | |
| | PF | 0.8500 | 0.9422 | | | | |
| 2 | Bus | 8 | 29 | 54.4 | 73.14 | 0.9856 | 1.0252 |
| | Rate (kW) | 1625.3 | 1625.3 | | | | |
| | PF | 0.9 | 0.9 | | | | |

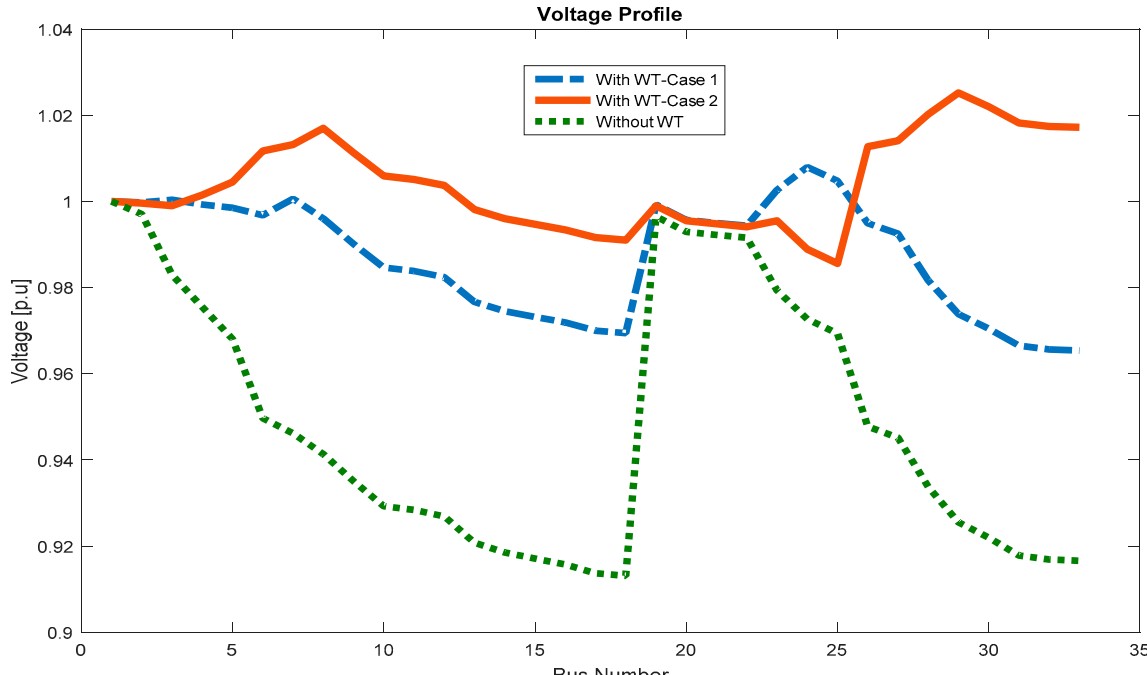

**Figure 8.** 32-bus grid voltage profile before and after installing the turbines with respect to constraints.

In order to demonstrate the superiority of the proposed PSO as multi-objective site selection method, the results of this method are compared with the results of Ref. [17] that PSO is applied for optimal site selection but as single-objective for 84 bus grid. In Table 7, the WTs site selection of proposed multi-objective method is compared with Reference [12] in two WTs optimal site selection without considering maximum allowable constraint. The results showed that the proposed method is obtained suitable results in view of loss reduction and also better minimum and maximum voltage values in comparison with Reference [17].

**Table 7.** Comparison results of the proposed method with Ref. [17], 84 bus grid.

| Number of WTG | WTG Placement | | | Loss | Percent Reduction of Loss | Minimum Voltage | Maximum Voltage |
|---|---|---|---|---|---|---|---|
| PSO Multi-objective | Bus | 8 | 81 | 368.4 | 30.72 | 0.9488 | 1.016 |
| | Rate (kW) | 5000 | 5000 | | | | |
| | PF | 0.9037 | 0.8872 | | | | |
| [17] | Bus | 7 | 80 | 368.3 | 30.7 | 0.9481 | 1.0136 |
| | Rate (kW) | 5000 | 5000 | | | | |
| | PF | 0.91 | 0.87 | | | | |

The results of the optimal site selection of wind turbines in the 84 and 32 bus distribution systems are presented below:

- The results of the multi-objective site selection of wind turbines are more rational than the single-objective results because of considering the both objective function and focus on loss and voltage profile.
- Site selection results considering the maximum capacity constraint of wind turbines in terms of losses and voltages are better than the one without considering the wind turbine constraint.
- Considering the maximum allowable capacity of wind turbines and the variable wind turbine capacity, this allows the program, in addition to optimal location, to determine the optimum wind turbine capacity according to the problem constraints to achieve the best objective function.

## 5. Conclusions

In this paper, optimal site selection of WTs while considering their maximum allowable capacity to reduce losses and improve voltage profile of distribution grids using the PSO was presented. This problem was solved as a multi-objective problem using weight coefficient method. The decision variables included the site, capacity, and optimal power factor of the WTs determined by the optimization method. The proposed method was implemented on 84- and 32-bus standard grids. In this study, a two-stage PSO algorithm was proposed to facilitate the site selection of WTs while considering their maximum capacity. The simulation results showed that this method is effective for fast and desirable site selection of WTs and can obtain the optimal position, maximum allowable capacity, and power factor for the WT. The results of this two-step new algorithm, including the improvement in the voltage profile and reduction in power losses, indicated the efficiency of this method. The results also demonstrated that the multi-objective problem solving resulted in a more realistic and accurate performance of the grid. In addition, the consideration of constraints has resulted in less loss and better voltage profile in comparison with not considering the WT capacity constraint. Additionally, considering this capacity constraint allows that the optimal capacity of WT is determined according to the problem constraints to achieve the best objective function.

**Author Contributions:** All authors have contributed equally to this work. All authors of this manuscript jointly have conceived the theoretical analysis, modeling, and obtained the simulation.

**Funding:** This research was funded by the Ministry of Education (MOE) and Universiti Teknologi Malaysia for the financial support under the Post-Doctoral Fellowship Scheme (grant numbers 4F828, 04E54, and 18H10).

**Conflicts of Interest:** The authors declare no conflict of interest.

## Nomenclature

| | | | |
|---|---|---|---|
| $P_{loss}$ | Power loss | | |
| $I_k$ | Current of k line | $L_{f,i}$ | Current of feeder $i$ |
| $N_{br}$ | Number of lines | $L_{f,i}^{max}$ | Maximum current of feeder $i$ |
| $R_k$ | Resistance of k line | $P_{min,w,i}$ | Minimum power of WTG$_i$ |
| $V_j$ | Voltage of j bus | $P_{max,w,i}$ | Maximum power of WTG$_i$ |
| $V_i$ | Voltage of i bus | $V_{min}$ | Minimum bus voltage |
| $Q_i$ | Injected reactive power to i bus | $V_{max}$ | Maximum bus voltage |
| $X_k$ | Reactance of k line | $pf_{min,i}$ | Minimum value of the power factor WTG $_i$ |
| $V_m$ | Voltage of bus | $pf_{max,i}$ | Maximum value of the power factor WTG $_i$ |
| $N_{bus}$ | Number of buses | $S$ | Position vector |
| $P_{WTG}$ | Power of wind turbine generator | $V$ | Velocity vector |
| $pf_{WTG}$ | Power factor of wind turbine generator | $P$ | Vector of best position of particles |
| $P_{ij}^{Line}$ | Power of the line between buses of i and j | $P_{gb}$ | Best position in the entire community |
| $P_{ij,max}^{Line}$ | Maximum power of the line between buses of i and j | $w$ | Factor of inertia |
| $P_i$ | Injected active power to i bus | $w_{min}$ | Minimum value of inertia factor |
| $Nf$ | Number of feeders | $w_{max}$ | Maximum value of inertia factor |

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
