# Peer review of "A Multi-Objective Optimization Problem for Optimal Site Selection of Wind Turbines for Reduce Losses and Improve Voltage Profile of Distribution Grids"

_energies, doi:10.3390/en12132621_

Round 1

Reviewer 1 Report

This study concerns a multi-objective optimization problem for site selection of wind turbine to reduce losses and improve the voltage profile of distribution grids, based on particle swarm optimization method. Before the final acceptance can be recommended, several issues should be solved by authors as follows:

1. Reviewers believe that there is little literature to support the research foundation of this paper. More relevance of the work needs to be supplemented appropriately in the Introduction section. Please pay attention to this problem. For determining site selection of wind turbines, there are many existing issues but authors have no mention in the context of earlier work in the literature. Thus, an updated and complete literature review should be conducted and should appear as part of the Introduction. Please provide relevant information and answer to basic questions such as: What problem was studied and why is it important? What methods were used? What are the important results? What is missing (i.e., research gaps)? What conclusions can be drawn from the results?

2. Abstract and Conclusion are the same, which is highly undesirable. Perhaps the authors misunderstood the way that the summary and conclusion were written. Please rewritten them. The former should discuss what has done and how, while the latter should present only the major ‘specific’ outcome of the study without any repetition of information. Especially, please include specific and quantitative results. Please pay attention to this problem.

3. Authors highlight a multi-objective problem, but there is no clear objective. Please declare it in the text. In my opinion, multi-objective factors include maximum allowable capacity, minimizing losses and improving the voltage profile of distribution grids.

4.  Line 77: some words in this sentence are missing. Line 135: for “C=and C2”, it should be an error.

5. Section 3.1, for the description of, please add the correlation between particle swarm optimization (PSO) method and this paper.

6. For PSO method, errors and uncertainties in the study are explicitly included.

7. In this study, the assumptions of the study should be detailed. The authors seem to ignore assumptions for PSO method.

8. Section 4, please give some specific and quantitative results from the Figures and Table to support the proposed methods and its background.

9. Please describe in the manuscript that how the authors validate the PSO method.

10. How can accurate to evaluate all parameters (maximum allowable capacity, minimizing losses and improving the voltage profile) in your promoted model.

11. Proofreading through the full text is necessary to improve the quality of the English language. 

Authors are encouraged to answer these questions in detail point by point, instead of answering "*** has been revised." Please give the revised text in "Response to Reviewer " to save the reviewer time and improve the review efficiency.  Thanks for your cooperation.

Author Response

Energies Journal

Authors’ response to the review of paper energies-531178

Paper title:      

Manuscript Number: 

A multi-objective   optimization problem for optimal Site Selection of Wind turbines for Reduce   Losses and Improve Voltage Profile of Distribution  Grids

energies-531178

Dear Prof,

   We would like to express our most sincere gratitude for your effort and patience in reviewing our manuscript. We deeply appreciate your constructive comments which greatly help us to improve the technical quality and the presentation of this manuscript. This paper has been revised according to the relevant comments. The following is provided to outline each change made (point by point) as raised in the reviewer comments. The corrected sentences in the manuscript are highlighted in yellow fonts.

We look forward to your reply.

Kind regards,

Reviewer I

1.       Reviewers believe that there is little literature to support the research foundation of this paper. More relevance of the work needs to be supplemented appropriately in the Introduction section. Please pay attention to this problem. For determining site selection of wind turbines, there are many existing issues but authors have no mention in the context of earlier work in the literature. Thus, an updated and complete literature review should be conducted and should appear as part of the Introduction. Please provide relevant information and answer to basic questions such as: What problem was studied and why is it important? What methods were used? What are the important results? What is missing (i.e., research gaps)? What conclusions can be drawn from the results?

Reply:

The literature review is modified as follows and insert into the paper text. (Line 52-92)

2.       Abstract and Conclusion are the same, which is highly undesirable. Perhaps the authors misunderstood the way that the summary and conclusion were written. Please rewritten them. The former should discuss what has done and how, while the latter should present only the major ‘specific’ outcome of the study without any repetition of information. Especially, please include specific and quantitative results. Please pay attention to this problem.

Reply:

The abstract is modified as follows and insert into the paper text. ( Line 19-33). Also, the conclusion is edited as following and insert to the paper text. (Line 370-373).

3.       3. Authors highlight a multi-objective problem, but there is no clear objective. Please declare it in the text. In my opinion, multi-objective factors include maximum allowable capacity, minimizing losses and improving the voltage profile of distribution grids.

Reply:

The objective function of the problem is loss and voltage deviations minimizing as a multi-objective problem. The objective function is formulated based on weighted coefficients method as F=W1(F1/F1,max)+W2(F2/ F2,max). The W1 (weighted coefficient of F1) and W2 (weighted coefficient of F2) are selected 0.6 and 0.4 (based on user experience) respectively. F1,max and F2,max is the maximum value of F1 and F2 respectively. The voltage profile improvement means voltage deviations reduction or minimization. (Line 108-111)

4.       4.  Line 77: some words in this sentence are missing. Line 135: for “C=and C2”, it should be an error.

Reply:

This text is edited as follow:

In line 77 (old) or 103 (new): Where Vm (n) is the bus of i voltage and Nbus is the number of buses of the distribution grid.

In line 135 (old) or 157 (new): C1 and C2 are the power factors between two community and individual forces.

5.       Section 3.1, for the description of, please add the correlation between particle swarm optimization (PSO) method and this paper.

Reply:

In this study the decision variables include of optimal size, site and power factor of WTs are determined using PSO that described in detail in 3.2 section. (line 185).

6.       For PSO method, errors and uncertainties in the study are explicitly included..

Reply:

In the objective function of this article, the uncertainty of production and load has not been considered and PSO parameters such as the number of population, maximum repetition and C1 and C2 parameters are selected based on trial and error. (Line 177-184)

7.       7. In this study, the assumptions of the study should be detailed. The authors seem to ignore assumptions for PSO method.

Reply:

The In this study, the assumptions of the problem are that the maximum size of each wind turbine for 84-bus grid is considered 5 MW and for 32 bus grid each wind turbine size is considered 2 MW. Operating constraints are also considered, and the minimum and maximum bus voltages are considered between 0.9 and 1.05 p.u. Also, the C1 and C2 parameters for the PSO are considered equal to 2, also w_max and w_min are equal to 0.9 and 0.4 which are selected based on the trial and error method.

This above text is added to the paper text in the end of section  3.1 (Line 134)

8.       8. Section 4, please give some specific and quantitative results from the Figures and Table to support the proposed methods and its background.

Reply:

The results of the optimal site selection of wind turbines in the 84 and 32 bus distribution systems are presented below: (Line 346-357)

• The results of the multi-objective site selection of wind turbines are more rational than the single-objective results because of considering both objective functions and focus on loss and voltage profile.

• Site selection results considering the maximum capacity constraint of wind turbines in terms of losses and voltages are better than the one without considering the wind turbine constraint.

• Considering the maximum allowable capacity of wind turbines and the variable wind turbine capacity, this allows the program, in addition to the optimal location, to determine the optimum wind turbine capacity according to the problem constraints to achieve the best objective function.

9.       9. Please describe in the manuscript that how the authors validate the PSO method.

Reply:

In order to demonstrate the superiority of the proposed PSO as multi-objective site selection method, the results of this method are compared with the results of Ref. [17] that PSO is applied for optimal site selection but as single-objective. In Table 7, the WTs site selection of the proposed multi-objective method is compared with [12] in two WTs optimal site selection without considering maximum allowable constraint. The results showed that the proposed method is obtained suitable results in view of loss reduction and also better minimum and maximum voltage values in comparison with [17]. Also achieving to the optimal site, size and power factor of WTs in all the obtained results of the paper showed the good performance of the PSO.  (Line 336-345)

10.    10. How can accurate to evaluate all parameters (maximum allowable capacity, minimizing losses and improving the voltage profile) in your promoted model.

Reply:

In this study, the assumptions of the problem are that the maximum size of each wind turbine for 84-bus grid is considered 5 MW and for 32 bus grid each wind turbine size is considered 2 MW. the results showed that the power loss of the grids with applying the WTs are reduced and also the voltage profile of the grids are improved. The validation of proposed PSO are presented in response to the authors in comment 9 as follows:

In order to demonstrate the superiority of the proposed PSO as multi-objective site selection method, the results of this method are compared with the results of Ref. [17] that PSO is applied for optimal site selection but as single-objective. In Table 7, the WTs site selection of the proposed multi-objective method is compared with [12] in two WTs optimal site selection without considering maximum allowable constraint. The results showed that the proposed method is obtained suitable results in view of loss reduction and also better minimum and maximum voltage values in comparison with [17]. Also achieving to the optimal site, size and power factor of WTs in all the obtained results of the paper showed the good performance of the PSO. (Line 346-342)

11.    11. Proofreading through the full text is necessary to improve the quality of the English language.

Reply:

Thank you for this comment. The English language of the paper will be proofread by Elsevier Editing Language after finish all revision.

Authors are encouraged to answer these questions in detail point by point, instead of answering "*** has been revised." Please give the revised text in "Response to Reviewer " to save the reviewer time and improve the review efficiency.  Thanks for your cooperation.

Thank you very much.

Sincerely Yours

Dr Amirreza Naderipour

Institute of High Voltage & High Current

School of Electrical Engineering

Universiti Teknologi Malaysia

81310 UTM, Johor Bahru

Johor, Malaysia

Tel : +607 5535366

Reviewer 2 Report

Dear Authors,

Thank you for the opportunity to review your manuscript. I found the presented manuscript interesting, fitting well the current research directions and delivering a good approach for the considered problem. What is more, the manuscript is also within the scope of Energies journal interests.

I think that the following issues should be considered to improve the manuscript:

The literature review section is relatively short and does not refer to similar works in a detailed manner. I suggest expanding this section to provide the broader context of your research. 

The discussion section although mentioned in the subsection title is not really a discussion. There you should clearly ask if what you have calculated/found out makes sense and how it refers to the existing body of knowledge. What are the differences and similarities? 

In the modeling section, it is very unclear to me how the wind turbines power output has been modeled. Is it based on wind speed? If so, did you assume a potential difference in wind speed at different parts of the standard grid (they may be some kilometers apart)? 

Author Response

Energies Journal

Authors’ response to the review of paper energies-531178

Paper title:      

Manuscript Number: 

A multi-objective   optimization problem for optimal Site Selection of Wind turbines for Reduce   Losses and Improve Voltage Profile of Distribution  Grids

energies-531178

Dear Prof,

   We would like to express our most sincere gratitude for your effort and patience in reviewing our manuscript. We deeply appreciate your constructive comments which greatly help us to improve the technical quality and the presentation of this manuscript. This paper has been revised according to the relevant comments. The following is provided to outline each change made (point by point) as raised in the reviewer comments. The corrected sentences in the manuscript are highlighted in green fonts.

We look forward to your reply.

Kind regards,

Reviewer II

1.       The authors mentioned "Power generation by a WT demonstrates advantages such as an increase in system reliability" may not be appropriate since wind power system brought quite a few stability issues to the grid.

Reply:

In this paper, the objective is optimizing the wind turbines in the distribution network with the aim of reducing losses and improve network voltage profile. Considering that the reliability improvement of the distribution system is not the goal of this paper, according to the Reviewer 2 comment, the mentioned sentence was modified as follow:

Power generation by a WT demonstrates advantages such as the reduction in grid power losses, improvement in voltage profile, improvement in peak traffic, and faster transportation on transmission and distribution lines [2]. (Line 45-47)

2.       The motivation for the proposed work is not clearly presented. There is not enough literature review for the state-of-the-art methods for wind turbine siting. This is not a new topic in terms of technologies and statements of the problem. Summary of the pros and cons of existing methods are important to highlight the contribution of this work.

Reply:

The literature review is modified as follows and insert into the paper text. (Line 52-87).

3.       Quite a lot of variables are not defined nor explained in the paper. A nomenclature is necessary to help readers understand the items and equations better.

Reply:

A nomenclature as following is added to the paper text. (Line 36)

4.       Line 77, "Where is the bus i voltage domain and Nbus is the number of buses of the distribution grid" is missing some words?

Reply:

This sentence is modified as follow:            

Where,  is the bus of i voltage and Nbus is the number of buses of the distribution grid. (Line 103).

5.       The introduction of PSO can be streamlined as it is already a well-known method.

Reply:

According to the reviewer comment, the introduction of PSO streamlined.

6.       A flowchart or algorithm table is recommended to better present the proposed procedures (3.2).

Reply:

The flowchart of the problem solution is shown in the text. (Line 214)

7.       The formulation of optimization is not clearly presented. Is the main objective to minimize the loss and maximize the voltage profile?

Reply:

The objective function of the problem is loss and voltage deviations minimizing as a multi-objective problem. The objective function is formulated based on weighted coefficients method as F=W1(F1/F1,max)+W2(F2/ F2,max). The W1 (weighted coefficient of F1) and W2 (weighted coefficient of F2) are selected 0.6 and 0.4 (based on user experience) respectively. F1,max and F2,max is the maximum value of F1 and F2 respectively. The voltage profile improvement means voltage deviations reduction or minimization. (Line 108-111).

8.     The contribution of the proposed method needs to be further clarified.

Reply:

This text is added to the end of the conclusion to clarify the obtained results. (Line 355-359)

Thank you very much.

Sincerely Yours

Dr Amirreza Naderipour

Institute of High Voltage & High Current

School of Electrical Engineering

Universiti Teknologi Malaysia

81310 UTM, Johor Bahru

Johor, Malaysia

Tel : +607 5535366

Reviewer 3 Report

This paper proposes an optimal siting method for wind turbines with consideration of their maximum allowable capacity to reduce losses and improve the voltage profile of distribution grids using particle swarm optimization. Case studies with 32 and 84 bus systems are carried out. The author has the following questions/comments:

1. The authors mentioned "Power generation by a WT demonstrates advantages such as an increase in system reliability" may not be appropriate since wind power system brought quite a few stability issues to the grid.

2. The motivation for the proposed work is not clearly presented. There is not enough literature review for the state-of-the-art methods for wind turbine siting. This is not a new topic in terms of technologies and statements of the problem. Summary of the pros and cons of existing methods are important to highlight the contribution of this work.

3. Quite a lot of variables are not defined nor explained in the paper. A nomenclature is necessary to help readers understand the items and equations better.

4. Line 77, "Where, is the bus i voltage domain and Nbus is the number of buses of the distribution grid" is missing some words?

5. The introduction of PSO can be streamlined as it is already a well-known method.

6. A flowchart or algorithm table is recommended to better present the proposed procedures (3.2).

7. The formulation of optimization is not clearly presented. Is the main objective to minimize the loss and maximize the voltage profile?

8. The contribution of the proposed method needs to be further clarified.

Author Response

Energies Journal

Authors’ response to the review of paper energies-531178

Paper title:      

Manuscript Number: 

A multi-objective   optimization problem for optimal Site Selection of Wind turbines for Reduce   Losses and Improve Voltage Profile of Distribution  Grids

energies-531178

Dear Prof,

   We would like to express our most sincere gratitude for your effort and patience in reviewing our manuscript. We deeply appreciate your constructive comments which greatly help us to improve the technical quality and the presentation of this manuscript. This paper has been revised according to the relevant comments. The following is provided to outline each change made (point by point) as raised in the reviewer comments. The corrected sentences in the manuscript are highlighted in yellow fonts.

We look forward to your reply.

Kind regards,

Reviewer III

1.       The literature review section is relatively short and does not refer to similar works in a detailed manner. I suggest expanding this section to provide the broader context of your research.

Reply:

The literature review is modified as follows and insert into the paper text.

(Line 52-87)

2.       The discussion section although mentioned in the subsection title is not really a discussion. There you should clearly ask if what you have calculated/found out makes sense and how it refers to the existing body of knowledge. What are the differences and similarities?

Reply:

In the section of Simulation results and discussion in sub-section of 4.1 for 84-Bus grid, the bellow cases are presented. (Line 237, 244,264,331-342).

4.1.1 Optimal site selection of turbines without regard to constraints

•              Results of solving the problem as single-objective

•              Results of solving the problem as multi-objective

4.1.2 Optimal site selection of turbines with respect to constraints

(In this section the results of solving the problem is presented as multi-objective )

Also in the 32 bus network, the results of the site selection were considered as multi-objective, considering the maximum allowable capacity of wind turbines.

The results of the determination of the site and size of wind turbines without and with regard to the maximum allowable capacity of wind turbines are presented. There is also a comparison between single and multi-objective results. In the multi-objective results section, multi-objective results are more rational results with both losses reduction and voltage profile improvement objectives. It has also been stated that considering the maximum allowable capacity of wind turbines, it more reduces the distribution network losses and improves the voltage as long as the capacity of the fixed wind turbines (maximum capacity capability of wind turbines) is considered. On the other hand, in Table 5, the results of the site and size selection of 1 to 4 wind turbines with and without considering the maximum allowable capacity of wind turbines for the 84 bus network are compared, which the losses and the minimum and maximum voltages in the state with regard to the maximum allowable capacity of turbines are better than the one without considering the wind turbines constraint.

The results of the optimal site selection of wind turbines in the 84 and 32 bus distribution systems are presented below (this text is added to the paper text):

• The results of the multi-objective site selection of wind turbines are more rational than the single-objective results because of considering both objective functions and focus on loss and voltage profile.

• Site selection results considering the maximum capacity constraint of wind turbines in terms of losses and voltages are better than the one without considering the wind turbine constraint.

• Considering the maximum allowable capacity of wind turbines and the variable wind turbine capacity, this allows the program, in addition to the optimal location, to determine the optimum wind turbine capacity according to the problem constraints to achieve the best objective function.

3.       In the modelling section, it is very unclear to me how the wind turbines power output has been modelled. Is it based on wind speed? If so, did you assume a potential difference in wind speed at different parts of the standard grid (they may be some kilometres apart)?

Reply:

Wind turbines generate power by receiving the wind speeds. In this study, it is assumed that maximum size of per wind turbine with receiving the wind speeds for 84 bus system, is 5 MW and for 32-bus system, is 2 MW, considering the wind speed of that region. Therefore, every wind turbine is considered in terms of its peak generation capacity. As an example, in the following reference, each wind turbine is intended for placement in the distribution system with a peak power of 3 MW. In the placement problem, the site, size, as well as the power factor of each wind turbine, is optimally determined by the optimization algorithm.

Nowdeh, S. A., Davoudkhani, I. F., Moghaddam, M. H., Najmi, E. S., Abdelaziz, A. Y., Ahmadi, A., ... & Gandoman, F. H. (2019). Fuzzy multi-objective placement of renewable energy sources in the distribution system with the objective of loss reduction and reliability improvement using a novel hybrid method. Applied Soft Computing, 77, 761-779.

This text is added to the paper text in section 4. (Line 219-225)

In this section, the results of optimal site selection of wind turbines in 84 and 32 bus distribution systems are presented as single and multi-objective optimization with and without the maximum allowable capacity of WTs using PSO algorithm. Wind turbines generate power by receiving the wind speeds. In this study, it is assumed that maximum size of per wind turbine with receiving the wind speeds for 84 bus system, is 5 MW and for 32-bus system, is 2 MW, considering the wind speed of that region. Therefore, every wind turbine is considered in terms of its peak generation capacity [20]. In the placement problem, the site, size, as well as the power factor of each wind turbine, is optimally determined by the optimization algorithm.

[20] Nowdeh, S. A., Davoudkhani, I. F., Moghaddam, M. H., Najmi, E. S., Abdelaziz, A. Y., Ahmadi, A., ... & Gandoman, F. H. (2019). Fuzzy multi-objective placement of renewable energy sources in the distribution system with the objective of loss reduction and reliability improvement using a novel hybrid method. Applied Soft Computing, 77, 761-779

Thank you very much.

Sincerely Yours

Dr Amirreza Naderipour

Institute of High Voltage & High Current

School of Electrical Engineering

Universiti Teknologi Malaysia

81310 UTM, Johor Bahru

Johor, Malaysia

Tel : +607 5535366

Round 2

Reviewer 1 Report

The authors have addressed all my comments, therefore the reviewer suggests to accept the paper in the present form.

Reviewer 2 Report

Dear Authors,

Thank you for the revision. Please consider ordering the nomenclature table in alphabetical order and adding the doi numbers to the references. 

Reviewer 3 Report

The authors have answered my questions. The reviewer has no further comments.

This manuscript is a resubmission of an earlier submission. The following is a list of the peer review reports and author responses from that submission.